# Land-Use and Land-Cover (LULC) Change Detection in Wami River Basin, Tanzania

**Sekela Twisa [1,2,]*** and **Manfred F. Buchroithner [2]**

[1]  Institute for Integrated Management of Material Fluxes and of Resources (UNU-FLORES),
    United Nations University, Ammonstrasse 74, 01067 Dresden, Germany

[2]  Institute for Cartography, Technische Universität Dresden, 01062 Dresden, Germany

*   Correspondence: twisa@unu.edu; Tel.: +49-351-8921-9370

**Abstract:** Anthropogenic activities have substantially changed natural landscapes, especially in regions which are extremely affected by population growth and climate change such as East African countries. Understanding the patterns of land-use and land-cover (LULC) change is important for efficient environmental management, including effective water management practice. Using remote sensing techniques and geographic information systems (GIS), this study focused on changes in LULC patterns of the upstream and downstream Wami River Basin over 16 years. Multitemporal satellite imagery of the Landsat series was used to map LULC changes and was divided into three stages (2000–2006, 2006–2011, and 2011–2016). The results for the change-detection analysis and the change matrix table from 2000 to 2016 show the extent of LULC changes occurring in different LULC classes, while most of the grassland, bushland, and woodland were intensively changed to cultivated land both upstream and downstream. These changes indicate that the increase of cultivated land was the result of population growth, especially downstream, while the primary socioeconomic activity remains agriculture both upstream and downstream. In general, net gain and net loss were observed downstream, which indicate that it was more affected compared to upstream. Hence, proper management of the basin, including land use planning, is required to avoid resources-use conflict between upstream and downstream users.

**Keywords:** LULC; remote sensing; geographical information system; change detection; upstream; downstream

## 1. Introduction

Land-use and land-cover (LULC) change has become a fundamental and essential component in current strategies for monitoring environmental changes and managing natural resources [1,2]. Increasing anthropogenic activities around the biosphere are causing large-scale alterations of the Earth's land surface, which affect the effectiveness of global systems [3]. LULC and its resources have been used for the social, material, cultural, and spiritual needs of humans, while, in the process, humans have caused significant changes [4]. The rapid changes of LULC, particularly in developing countries [5,6], have resulted in the reduction of different vital resources including water, soil, and vegetation [7–9]. Furthermore, the actions leading to LULC changes have a local cause. However, because of their speed, extension, and intensity, they have numerous and critical global implications [10], particularly on natural resources. The increasing change is alarming and can significantly impact the local, regional, national, and worldwide environment [11,12].

Researchers, policymakers, and planners utilize LULC information to determine changes in natural resources, including evaluating growth patterns [13]. A better understanding of land dynamics requires the detection of LULC change [14]. Empirical studies by researchers from diverse disciplines

proved that changes in LULC are key to various applications such as hydrology, agriculture, forest, environment, geology, and ecology [15,16]. Studies on LULC change detection have always attracted the attention of scientists [12,17,18]. Many researchers argue that LULC change could result in ecosystem imbalance and impacts on the environment caused by humans and their role in climate change [9,19–22]. Furthermore, their findings yield arguments that focus on interventions, which require evidence on the impacts and rates of LULC change and the distribution of these changes in time and space as a fundamental factor in present strategies for environmental monitoring of changes and managing natural resources [23].

Several studies have proven the effectiveness of space-borne imagery to monitor LULC changes [24–26], specifically in Africa. In 2010, El Gammal et al. [27] used numerous Landsat images of different periods (1972, 1982, 1987, 2000, 2003, and 2008) to analyze the changes of the shores of Lake Nasser, Egypt, and the river volume. In 2011, Mahmoud et al. [28] tested a field-based approach for classifying land cover using TerraSAR-X imagery. Akinyemi [29] used Landsat images from different years (1987, 2003, and 2016) to study LULC change in the central Albertine Rift in northwestern Rwanda. Cheruto et al. [9] assessed LULC changes using geographic information systems (GIS) and remote sensing techniques in Makueni County, Kenya. The LULC classification is perhaps among the most well-known applications of geospatial application [28,30].

Remote sensing (RS) and GIS have long been recognized as essential and powerful tools in determining LULC changes at different spatial scales [31]. Several change detection techniques and image analyses have been used to extract evidence from remotely sensed data [32,33]. GIS, on the other hand, integrate information derived from RS to develop a clear understanding of LULC modeling [34]. RS and GIS have proved to be very useful for the detection of LULC patterns [32,35–38]. Moreover, the combined use of satellite RS and GIS has proved to be a robust and cost-effective method for monitoring LULC changes [39–45]. With the development of RS and GIS techniques, LULC mapping has become a detailed and useful way to advance the choice of areas for different uses [1,14,46].

In developing countries, the resource bases, such as land, forest, and water, are declining significantly. However, information regarding the rate of reduction is often lacking [47]. Accurate LULC change information is necessary to understand the leading causes and the environmental costs of such changes. Moreover, analyzing the driving forces causing LULC change is essential to understanding the current changes to forecast future alterations. A study of LULC changing aspects and its driving forces in space and time provided the basis for the sustainability of natural resource systems because it was used to reflect the state of the watershed. Regardless of the increasing concerns about the impacts of LULC change on global changes of the environment and sustainable development [48], research on LULC change in Tanzania is minimal. In the Wami River Basin, the magnitude and dynamics of these changes have not been broadly studied. Little is known about the spatiotemporal extents of LULC change, and no information has been evaluated over time to improve land use planning in the basin. Moreover, to understand the aspects of changes in the human environment across space and time, numerous studies are required [49]. To address this, the integrated approach of RS and GIS data was used for monitoring the LULC change [50] in the Wami River Basin, both upstream and downstream. The outcome of this study is expected to be highly useful to planners, resource managers, and policymakers for sustainable use of resources in the basin.

## 2. Materials and Methods

### 2.1. Study Area

The Wami River Basin (Figure 1) is located between 5–7° S and 36–39° E. It covers the semi-arid areas in central Tanzania through the humid inland swamps in east-central Tanzania to the Indian Ocean and spans an area of 41,167 km$^2$. The relief of the Wami River Basin ranges from 2 m to 2370 m above sea level. The average rainfall in the basin ranges from 550 mm to 1000 mm per annum. There are two rainfall regions in the basin [51]:

- The unimodal rainfall region covers the western and southwestern parts (one wet period November-December-January-February-March-April (NDJFMA)) and
- The bimodal rainfall region includes the eastern and northeastern part of the basin (two wet periods, October-November-December (OND), and March-April-May (MAM)).

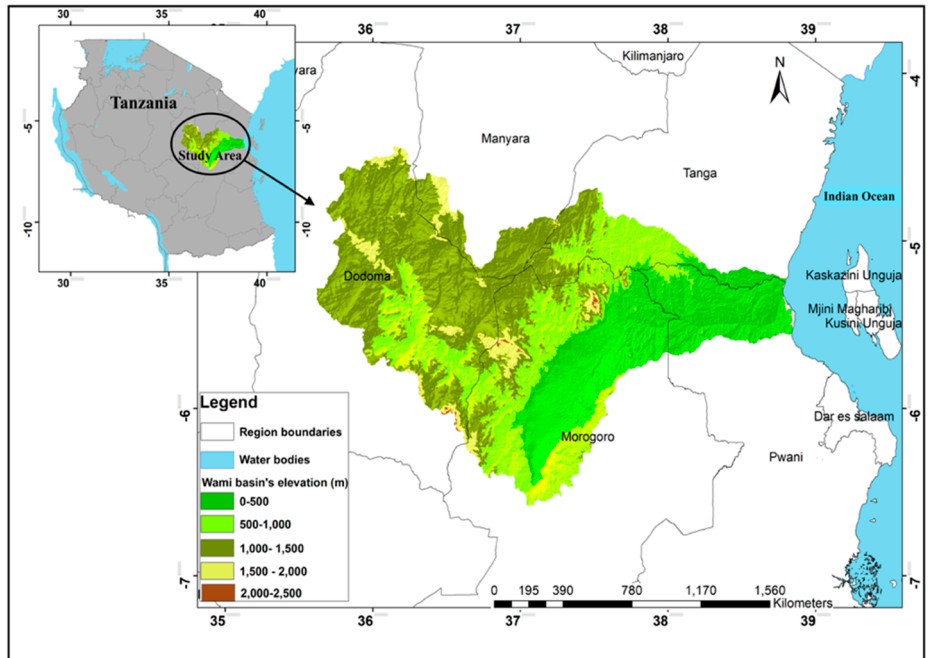

**Figure 1.** The study area of Wami River Basin.

The population of Wami/Ruvu Basin can be estimated to be 9.9 million based on the 2012 national population census. According to the inventory survey conducted within a project carried out by the Japan International Cooperation Agency (JICA, Tokyo, Japan) [52], the total population in Wami/Ruvu Basin in 2035 is forecasted to amount to 12.58 million, of which 7.39 million (59%) and 5.20 million (41%) are urban and rural populations, respectively.

## 2.2. Data Acquisitions and Preparation

Landsat imagery was used to determine LULC change, while dates were selected based on data quality, data availability, and the dry season (Table 1). Four Landsat imageries were acquired for the years 2000, 2006/2007, 2011, and 2016 from the USGS Global Visualization Viewer (https://glovis.usgs.gov) using Path/Row 168/64, 168/65 for Kinyasungwe Sub-catchment, and 167/64 and 166/64 for Wami Sub-catchment (Table 1). These data were used for the generation of LULC maps fed into on a GIS. The software packages, ERDAS Imagine 2011, Arc GIS 10.3, and QGIS 2.18 were employed at various stages of analysis. All the images have the same spatial resolution (30 m), but data were sensed by different sensors and satellites at different times of the year. Hence, each scene was radiometrically corrected by changing the raw digital numbers (DNs) into the top of atmosphere (TOA) reflectance values to correct for varying sun angles and changes in surface reflectance. After mosaicking the Landsat scenes by date, the resulting four images were clipped to the study area.

**Table 1.** Detailed data on Landsat images used in the study.

| Year | Satellite | Sensor | Path/Row | Resolution (m) | Acquisition Date | Cloud Cover |
|---|---|---|---|---|---|---|
| | Landsat 5 | TM | 166/64 | 30 | 8 July 2000 | 17% (not in the study area) |
| | Landsat 7 | ETM | 167/64 | 30 | 7 July 2000 | 5% |
| 2000 | Landsat 7 | ETM | 167/65 | 30 | 7 July 2000 | 2% |
| | Landsat 7 | ETM | 168/64 | 30 | 16 September 2000 | 0% |
| | Landsat 7 | ETM | 168/65 | 30 | 16 September 2000 | 7% |
| | Landsat 7 | ETM | 166/64 | 30 | 23 February 2006 | 5% |
| | Landsat 5 | TM | 167/64 | 30 | 24 January 2007 | 2% |
| 2006/2007 | Landsat 5 | TM | 167/65 | 30 | 24 January 2007 | 5% |
| | Landsat 7 | ETM | 168/64 | 30 | 17 September 2006 | 0% |
| | Landsat 7 | ETM | 168/65 | 30 | 17 September 2006 | 3% |
| | Landsat 5 | TM | 166/64 | 30 | 7 July 2011 | 10% (not in the study area) |
| | Landsat 7 | ETM | 167/64 | 30 | 26 October 2011 | 4% |
| 2011 | Landsat 7 | ETM | 167/65 | 30 | 6 July 2011 | 1% |
| | Landsat 7 | ETM | 168/64 | 30 | 11 June 2011 | 11% (not in the study area) |
| | Landsat 7 | ETM | 168/65 | 30 | 5 July 2011 | 5% |
| | Landsat 8 | OLI | 166/64 | 30 | 4 July 2016 | 10% (not in the study area) |
| | Landsat 8 | OLI | 167/64 | 30 | 24 May 2016 | 6% |
| 2016 | Landsat 8 | OLI | 167/65 | 30 | 24 May 2016 | 4% |
| | Landsat 8 | OLI | 168/64 | 30 | 16 June 2016 | 0% |
| | Landsat 8 | OLI | 168/65 | 30 | 16 June 2016 | 1% |

### 2.3. Classification and Change Detection

A hybrid classification technique was adopted to digitally categorize each Landsat image, since this technique has been shown to perform better in the case of spectral variability of individual cover types [53]. Several studies have suggested that hybrid classification produces superior results compared to unsupervised or supervised classification alone [54,55]. All the images were studied by assigning per-pixel signatures and differentiating the catchment into 10 classes by the specific DN of different landscape elements. The delineated classes were bushland, woodland, swamp, cultivated land, settlement area, grassland, water, forest, open land, and airfield (Table 2). For each of the predetermined LULC types, training samples were selected by delimiting polygons around representative sites. A suitable spectral signature is the one that ensures that there is "minimal confusion" among the land covers to be mapped [56]. A total of 60 spectral signatures for the respective LULC types derived from the satellite imagery were recorded using the pixels enclosed by these polygons.

**Table 2.** Land-use and land-cover (LULC) classification scheme.

| Class | Descriptions |
|---|---|
| Bushland | Mainly comprised of plants that are multi-stemmed from a single root base. |
| Woodland | An assemblage of trees with canopy ranging from 20% to 80% but which may, or rare occasions, is closed entirely. |
| Swamp | The low-lying, uncultivated ground where water collects; a bog or marsh. |
| Cultivated land | Crop fields and fallow lands. |
| Settlement area | Residential, commercial, industry, transportation, roads, mixed urban. |
| Grassland | Mainly composed of grass. |
| Forest | The continuous stand of trees, many of which may attain a height of 50 m including natural forest, mangrove and plantation forest. |
| Water | River, open water, lakes, ponds and reservoirs. |
| Open land | The land area of exposed soil and barren area influenced by a human. |
| Airfield | Area of the plot set aside for the take-off, landing and maintenance of aircraft. |

Afterward, the maximum likelihood classification (MCL) algorithm was used for the supervised classification of the images. This is the type of image classification that is mainly controlled by the analyst by selecting the pixels that are representative of the identified classes [57]. The results of the available higher resolution images and the hybrid classification comparison results revealed some degree of misclassification. Post-classification refinement was used to improve classification accuracy for the simplicity and efficiency of the method [58], which resulted in a cross-tabulation (transition) matrix. The LULC change-transition matrix was computed using the overlay procedure in ArcGIS to quantify the area converted from a particular LULC class to another LULC category during the study period. Later, visual interpretation was used to address the mixed pixels problem. Visual analysis reference data and local knowledge significantly enhanced the results obtained using the supervised algorithm. For the enhancement of classification accuracy and the quality of the LULC maps produced, the visual interpretation was critical. This approach was repeated for all incorrectly classified pixels, as determined from topographic maps and high-resolution satellite images.

Accuracy assessment is essential for individual classification if the classification data is to be useful in change detection [59]. Normally, classification accuracy is carried out by comparison of two datasets: One based on reference information, referred to as "ground truth", and the other being the result of the analysis of remotely sensed data [60]. For the accuracy assessment of LULC maps extracted from satellite images, the stratified random method for each of four LULC maps was used to represent the different LULC class of the study area. The accuracy was assessed using 60 pixels per category and was based on visual interpretation and ground truth data. A cross-tabulation was executed between the class values and the ground truth and presented the results as an error matrix. Moreover, the non-parametric Kappa test was performed to measure the extent of the classification accuracy to account for diagonal elements and elements in the confusion matrix [61].

## 3. Results

### 3.1. Accuracy Assessment

Overall, LULC classification levels for the four dates ranged from 91% to 97%, with Kappa indices of agreement ranging from 0.88 to 0.95. Accuracies per individual LULC class (i.e., user accuracy (UA) and producer accuracy (PA)) are shown in Tables 3 and 4. These values of Kappa for the four classification results are satisfactory for the study area because they satisfy the minimum 85% accuracy, as stipulated by Anderson's [62] classification scheme. These results provide a major platform for the subsequent analysis of LULC changes.

**Table 3.** Accuracy assessment of the LULC classification at Kinyasungwe Sub-catchment.

| LULC | 2000 | | 2006 | | 2011 | | 2016 | |
|---|---|---|---|---|---|---|---|---|
| | PA | UA | PA | UA | PA | UA | PA | UA |
| Bushland | 99.53 | 95.33 | 100 | 93.26 | 99.10 | 93.41 | 99.28 | 76.56 |
| Woodland | 98.87 | 98.94 | 99.85 | 96.35 | 100 | 96.62 | 98.15 | 91.89 |
| Swamp | 90.33 | 93.61 | 86.06 | 90.42 | 92.29 | 90.39 | 77.54 | 81.21 |
| Cultivated land | 70.02 | 95.41 | 54.82 | 97.01 | 76.30 | 97.40 | 76.32 | 96.42 |
| Settlement area | 77.87 | 100 | 84.62 | 100 | 90.39 | 100 | 50.45 | 100 |
| Grassland | 97.61 | 98.54 | 100 | 95.63 | 98.71 | 95.96 | 98.31 | 78.65 |
| Water | 70.15 | 95.67 | 91.17 | 98.56 | 41.72 | 100 | 32.95 | 95.01 |
| Forest | 100 | 87.10 | 100 | 93.88 | 100 | 93.87 | 100 | 50.97 |
| Open land | 100 | 100 | 100 | 100 | 100 | 100 | 100 | 100 |
| Overall | 97.81 | | 96.93 | | 97.26 | | 91.69 | |
| Kappa | 0.90 | | 0.88 | | 0.93 | | 0.95 | |

**Table 4.** Accuracy assessment of the LULC classification at Wami Sub-catchment.

|  | 2000 |  | 2006 |  | 2011 |  | 2016 |  |
| --- | --- | --- | --- | --- | --- | --- | --- | --- |
| **LULC** | **PA** | **UA** | **PA** | **UA** | **PA** | **UA** | **PA** | **UA** |
| Bushland | 100 | 80.25 | 100 | 84.71 | 100 | 80.65 | 100 | 72.87 |
| Woodland | 93.89 | 92.61 | 97.50 | 96.93 | 86.95 | 93.80 | 85.84 | 89.74 |
| Swamp | 95.49 | 90.21 | 99.37 | 95.26 | 97.16 | 89.91 | 96.13 | 8126 |
| Cultivated land | 97.07 | 93.91 | 100 | 97.51 | 95.36 | 97.78 | 94.72 | 81.59 |
| Settlement area | 100 | 77.87 | 100 | 87.60 | 100 | 64.68 | 100 | 60.73 |
| Grassland | 74.42 | 85.81 | 79.78 | 100 | 63.24 | 99.65 | 71.44 | 97.49 |
| Water | 70.01 | 100 | 68.22 | 100 | 34.92 | 100.22 | 11.82 | 100. |
| Forest | 100 | 100 | 100 | 100 | 100 | 100 | 100 | 100 |
| Open land | 85.68 | 89.74 | 81.38 | 86.37 | 79.96 | 89.34 | 76.93 | 90.65 |
| Overall | 97.33 |  | 97.67 |  | 94.92 |  | 91.25 |  |
| Kappa | 0.88 |  | 0.89 |  | 0.79 |  | 0.76 |  |

## 3.2. Upstream Sub-Catchment (Kinyasungwe)

Tables 5 and 6, along with Figure 2, summarize the trend of LULC change from 2000 to 2016 based on 10 classes extracted from Kinyasungwe Sub-catchment with a proportionate coverage area for each. Moreover, the spatial representation of LULC types from 2010 to 2016 is shown in Figure 3 Initially, when the study began in the year 2000, the pattern of LULC as the percentage of the total area studied was dominated by bushland, covering 33.67% of the total studied area, followed by woodland (30.2%), grassland (22.12%), cultivated land (8.4%), swamp (4.24%), forest (0.68%), settlement area (0.49%), water (0.18%), open land (0.01%), and airfield (0.01%). Furthermore, trend changes were observed for all LULC in years 2006, 2011, and 2016, except for water, open land, and airfield. In 2016, when the study ended, the observed LULC pattern was dominated by woodland at 28.07% followed by bushland at 25.77%, cultivated land at 22.45%, grassland at 17.69%, swamp at 4.5%, settlement area at 1.01%, and forest at 0.34% (Table 5). Furthermore, in Table 6, the pattern of LULC during the studied period (2000–2016) indicates the general decrease of natural areas, including bushland (7.9%), woodland (2.13%), forest (0.34%), and water (0.03%). The swamp area (0.26%) was the only natural area which increased in size. The land uses supporting the population growth and economic activities increased as indicated by settlements (0.52%) and cultivated land (14.05%).

**Table 5.** Results of the LULC classification for 2000, 2006, 2011, and 2016 images showing the area of each category and category percentages (Kinyasungwe Sub-catchment).

| Year | 2000 |  | 2006 |  | 2011 |  | 2016 |  |
| --- | --- | --- | --- | --- | --- | --- | --- | --- |
| **LULC (Area)** | **Area (km²)** | **%** | **Area (km²)** | **%** | **Area (km²)** | **%** | **Area (km²)** | **%** |
| Bushland | 5643.63 | 33.67 | 4940.49 | 29.47 | 4646.94 | 27.72 | 4320.24 | 25.77 |
| Woodland | 5062.27 | 30.2 | 4934.55 | 29.44 | 4770.86 | 28.46 | 4705.63 | 28.07 |
| Swamp | 710.45 | 4.24 | 796.38 | 4.75 | 775.95 | 4.63 | 754.23 | 4.50 |
| Cultivated land | 1408.16 | 8.4 | 2621.27 | 15.64 | 3210.88 | 19.16 | 3763.79 | 22.45 |
| Settlement area | 82.43 | 0.49 | 97.77 | 0.58 | 108.34 | 0.65 | 169.11 | 1.01 |
| Grassland | 3708.05 | 22.12 | 3279.77 | 19.57 | 3168.52 | 18.9 | 2965.09 | 17.69 |
| Water | 30.36 | 0.18 | 20.53 | 0.12 | 13.88 | 0.08 | 24.96 | 0.15 |
| Forest | 114.58 | 0.68 | 69.18 | 0.41 | 64.58 | 0.39 | 56.89 | 0.34 |
| Open land | 1.35 | 0.01 | 1.35 | 0.01 | 1.35 | 0.01 | 1.35 | 0.01 |
| Airfield | 0.39 | 0.01 | 0.39 | 0.001 | 0.39 | 0.001 | 0.39 | 0.01 |
| Total | 16,762 | 100 | 16,762 | 100 | 16,762 | 100 | 16,762 | 100 |

**Table 6.** Results of the LULC classification for 2000, 2006, 2011, and 2016 images showing the area changed and percentage at Kinyasungwe Sub-catchment.

| Year | 2000–2006 | | 2006–2011 | | 2011–2016 | | 2000–2016 | |
|---|---|---|---|---|---|---|---|---|
| LULC Change | Area (km$^2$) | % | Area (km$^2$) | % | Area (km$^2$) | % | Area (km$^2$) | % |
| Bushland | −703.14 | −4.2 | −293.55 | −1.75 | −326.7 | −1.95 | −1323.39 | −7.9 |
| Woodland | −127.72 | −0.76 | −163.69 | −0.98 | −65.23 | −0.39 | −356.64 | −2.13 |
| Swamp | +85.93 | +0.51 | −20.43 | −0.12 | −21.72 | −0.13 | 43.78 | 0.26 |
| Cultivated land | +1213.11 | +7.24 | +589.61 | +3.52 | +552.91 | +3.29 | 2355.63 | 14.05 |
| Settlement area | +15.34 | +0.09 | +10.57 | +0.07 | +60.77 | +0.36 | 86.68 | 0.52 |
| Grassland | −428.28 | −2.55 | −111.25 | +0.67 | −203.43 | −1.21 | −742.96 | −4.43 |
| Water | −9.83 | −0.06 | −6.65 | −0.04 | +11.08 | +0.07 | −5.4 | −0.03 |
| Forest | −45.4 | −0.27 | −4.6 | −0.02 | −7.69 | −0.05 | −57.69 | −0.34 |

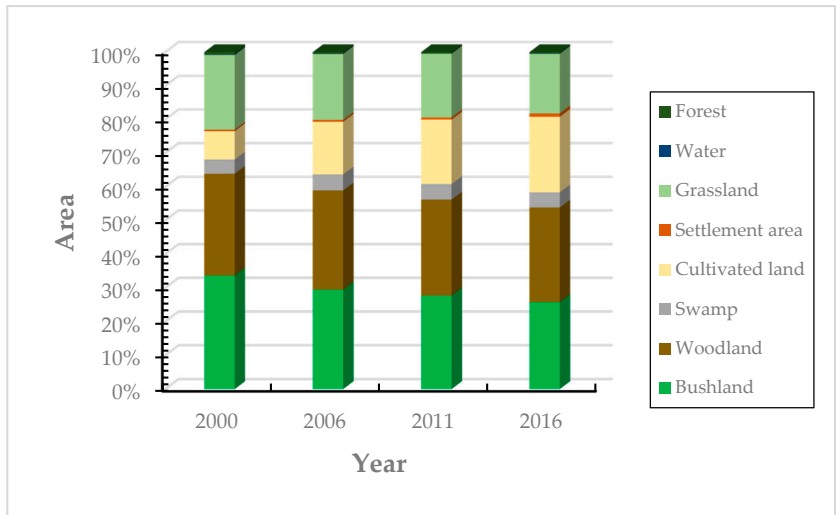

**Figure 2.** LULC change graph for 2000, 2006, 2011, and 2016 at Kinyasungwe Sub-catchment.

Table 7 indicates the areas changed with their corresponding percentages based on the change matrix cross-tabulation from 2000 to 2016. In the table, the LULC class is compared to another in terms of total area LULC class. During the study duration, no changes of settlement area were observed, as 100% remained intact, followed by cultivated land at 96%, woodland at 91%, swamp at 81%, grassland at 78%, bushland at 76%, forest at 50%, and water at 32%. Therefore, the highest conversion was experienced in the water area, as almost 65% of the total area was converted to swamp and the rest was converted to cultivated land (2%) and woodland (1%). Most of the forest was converted to woodland (42%) and grassland (5%). Although the cultivated land did not change much, almost 23% was gained from bushland (1298.03 km$^2$) followed by grassland (667.45 km$^2$), woodland (404.98 km$^2$), swamp (42.63), forest (1.15 km$^2$), and water (0.60 km$^2$). Between 2000 and 2016, swamp and forest were mainly converted to grassland, while water and grassland were converted to swamp, respectively. Relatively greater modifications took place from forest and swamp to woodland.

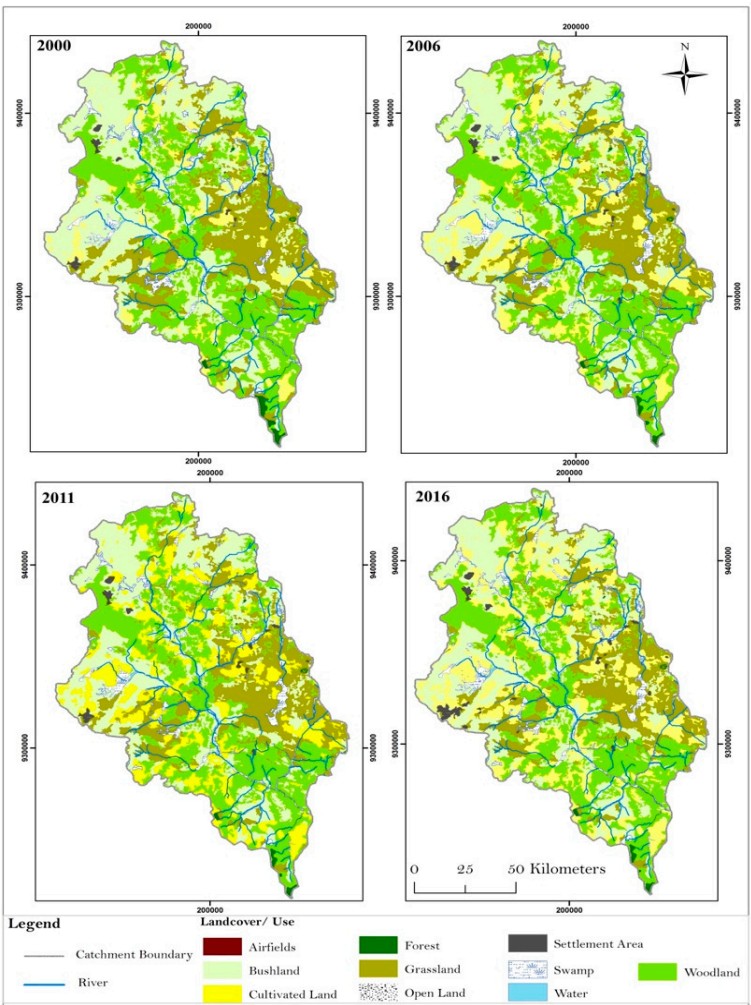

**Figure 3.** LULC maps for 2000, 2006, 2011, and 2016 at Kinyasungwe Sub-catchment (upstream).

**Table 7.** Transition matrix showing LULC change in Kinyasungwe Sub-catchment between 2000–2016.

| LULC | Unit | BL | WL | SWP | CL | SA | GL | WT | FR |
|---|---|---|---|---|---|---|---|---|---|
| BL | (km²) | 4289.16 | 0 | 56.43 | 1298.03 | 0 | 0 | 0 | 0 |
| | % | 76 | 0 | 1 | 23 | 0 | 0 | 0 | 0 |
| WL | (km²) | 0 | 4606.67 | 50.63 | 404.98 | 0 | 0 | 0 | 0 |
| | % | 0 | 91 | 1 | 8 | 0 | 0 | 0 | 0 |
| SWP | (km²) | 7.10 | 28.42 | 575.46 | 42.63 | 0 | 49.73 | 7.10 | 0 |
| | % | 1 | 4 | 81 | 6 | 0 | 7 | 1 | 0 |
| CL | (km²) | 14.08 | 0 | 14.08 | 1351.83 | 28.16 | 0 | 0 | 0 |
| | % | 1 | 0 | 1 | 96 | 2 | 0 | 0 | 0 |
| SA | (km²) | 0 | 0 | 0 | 0 | 82.43 | 0 | 0 | 0 |
| | % | 0 | 0 | 0 | 0 | 100 | 0 | 0 | 0 |
| GL | (km²) | 0 | 0 | 74.16 | 667.45 | 37.08 | 2892.28 | 0 | 0 |
| | % | 0 | 0 | 2 | 18 | 1 | 78 | 0 | 0 |
| WT | (km²) | 0 | 0.30 | 19.73 | 0.60 | 0 | 0 | 9.72 | 0 |
| | % | 0 | 1 | 65 | 2 | 0 | 0 | 32 | 0 |
| FR | (km²) | 2.29 | 48.12 | 0 | 1.15 | 0 | 5.73 | 0 | 57.29 |
| | % | 2 | 42 | 0 | 1 | 0 | 5 | 0 | 50 |

BL—Bushland, WL—Woodland, SP—Swamp, CL—Cultivated land, SA—Settlement area, GL—Grassland, WT—Water, FR—Forest. Note: The bold numbers on the diagonal represent unchanged LULC proportions from 2000 to 2016 and their corresponding percentages, while the others are the areas changed from one class to another.

### 3.3. Downstream Sub-Catchment (Wami)

Tables 8 and 9, together with Figure 4, summarize the trend of LULC change from 2000 to 2016 based on nine classes extracted from Wami Sub-catchment with a proportionate coverage area for each. The spatial representation of LULC types from 2010 to 2016 is shown in Figure 5. Initially, when the study began in the year 2000, the pattern of LULC as a percentage of the total area studied was dominated by woodland (61.86%), followed by bushland (16.21%), cultivated land (8.17%), forest (7.21%), grassland (4.31%), swamp (1.99%), water (0.16%), settlement area (0.05%), and open land (0.01%). The trend changes were observed for all land uses in the years 2006, 2011, and 2016, except for woodland, swamp, grassland, forest, and open land. In 2016, when the study ended, the observed LULC pattern as a percentage of the total area studied was dominated by woodland (52.10%), followed by cultivated land (25.36%), bushland (11.69%), forest (4.36%), grassland (3.69%), swamp (2.09%), settlement area (0.48%), water (0.18%), and open land (0.01%) (Table 8). During the study period (2000–2016), woodland decreased from 61.86% to 52.10%, bushland from 16.21% to 11.69%, grassland from 4.31% to 3.69%, and forest from 7.21% to 4.36%. However, these alteration changes did not occur equally for all land uses. The cultivated land experienced a significant increase from 8.17% to 25.36%, as well as in settlements from 0.05% to 0.48%, swamp from 1.99% to 2.09%, and water from 0.16% to 0.18% (Table 8). The results reveal that the highest net gain was in cultivated land (17.19%), followed by settlement area (0.43%), swamp (0.1%), and water (0.02%), while net loss was in woodland (9.76%), bushland (4.52%), forest (2.85%), and grassland (0.62%) (Table 9).

**Table 8.** Results of the LULC classification for 2000, 2006, 2011, and 2016 images showing the area of each category and category percentages (Wami Sub-catchment).

| Year | 2000 | | 2006 | | 2011 | | 2016 | |
|---|---|---|---|---|---|---|---|---|
| LULC | Area (km$^2$) | % | Area (km$^2$) | % | Area (km$^2$) | % | Area (km$^2$) | % |
| Bushland | 2336.81 | 16.21 | 1970.88 | 13.67 | 1870.67 | 12.98 | 1685.75 | 11.69 |
| Woodland | 8917.01 | 61.86 | 8621.68 | 59.81 | 8208.92 | 56.94 | 7511.02 | 52.10 |
| Swamp | 287.36 | 1.99 | 285.53 | 1.98 | 310.24 | 2.15 | 301.23 | 2.09 |
| Cultivated land | 1178.31 | 8.17 | 1989.03 | 13.8 | 2679.33 | 18.59 | 3655.96 | 25.36 |
| Settlement area | 7.71 | 0.05 | 11.28 | 0.08 | 22.73 | 0.16 | 68.86 | 0.48 |
| Grassland | 621.12 | 4.31 | 604.55 | 4.19 | 632.79 | 4.39 | 532.34 | 3.69 |
| Water | 22 | 0.16 | 23.34 | 0.17 | 24.74 | 0.17 | 26.05 | 0.18 |
| Forest | 1038.78 | 7.21 | 902.8 | 6.26 | 660.14 | 4.58 | 627.88 | 4.36 |
| Open land | 6.32 | 0.04 | 6.32 | 0.04 | 6.32 | 0.04 | 6.32 | 0.01 |
| Total | 14,415 | 100 | 14,415 | 100 | 14,415 | 100 | 14,415 | 100 |

**Table 9.** Results of the LULC classification for 2000, 2006, 2011, and 2016 images showing the area changed and percentage at Wami Sub-catchment.

| Year | 2000–2006 | | 2006–2011 | | 2011–2016 | | 2000–2016 | |
|---|---|---|---|---|---|---|---|---|
| LULC Change | Area (km$^2$) | % | Area (km$^2$) | % | Area (km$^2$) | % | Area (km$^2$) | % |
| Bushland | −365.93 | −2.54 | −100 | −0.69 | −184.92 | −1.29 | −651.06 | −4.52 |
| Woodland | −295.33 | −2.05 | −412.76 | −2.87 | −697.90 | −4.84 | −1405.99 | −9.76 |
| Swamp | −1.83 | −0.01 | +24.71 | +0.17 | −9.01 | −0.06 | +13.87 | +0.1 |
| Cultivated land | +810.72 | +5.63 | +690.30 | +4.79 | +977.09 | +6.77 | +2477.65 | +17.19 |
| Settlement area | +3.57 | +0.03 | +11.45 | +0.08 | +46.13 | +0.33 | +61.15 | +0.43 |
| Grassland | −16.57 | −0.12 | +28.24 | +0.20 | −100.45 | −0.70 | −88.78 | −0.62 |
| Water | +1.34 | +0.01 | +1.40 | +0.01 | +1.31 | +0.01 | +4.05 | +0.02 |
| Forest | −135.98 | −0.95 | −242.66 | −1.68 | −32.26 | −0.22 | −410.9 | −2.85 |

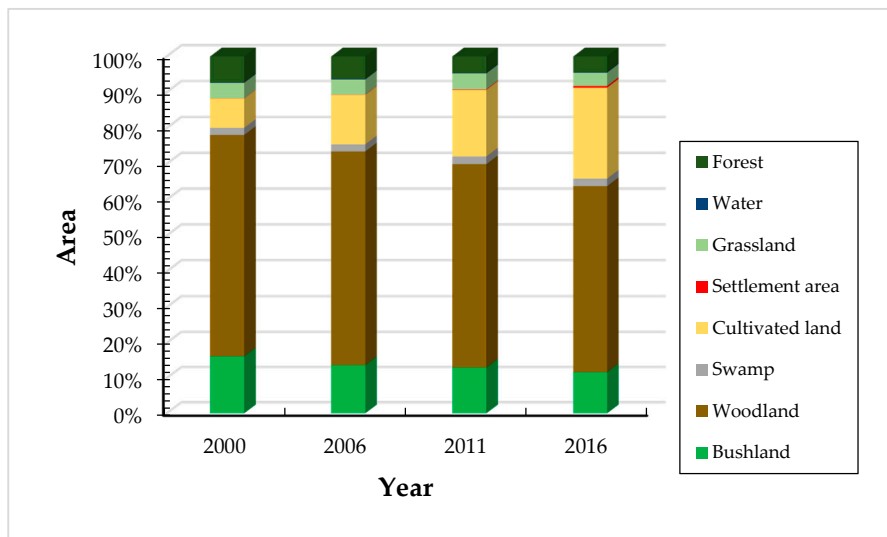

**Figure 4.** LULC change graph for 2000, 2006, 2011, and 2016 at Wami Sub-catchment.

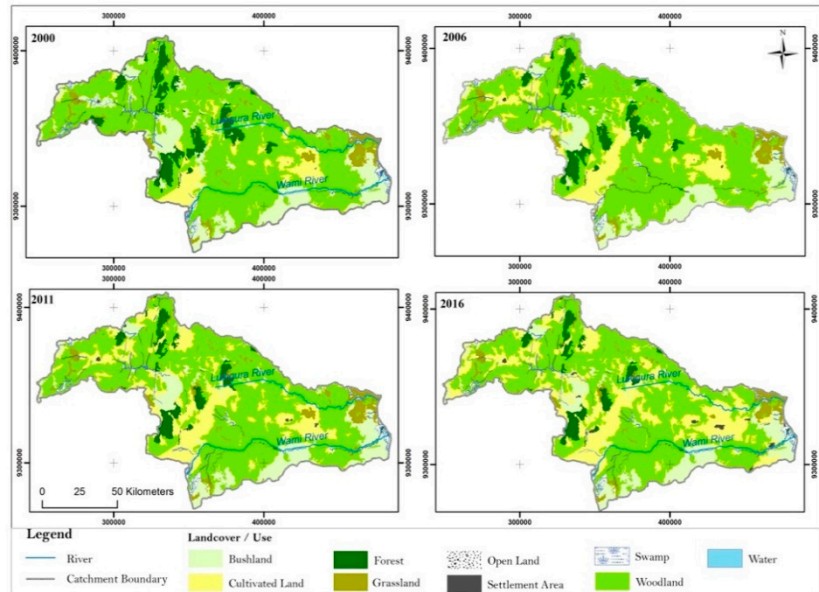

**Figure 5.** LULC maps for 2000, 2006, 2011, and 2016 at Wami Sub-catchment (downstream).

Table 10 shows the cross-tabulation change matrix for the changed areas and their corresponding percentages from one LULC class to another in comparison with the total area of each LULC class from 2000 to 2016. During the study period, 100% of settlement area remained unchanged, followed by cultivated land (97%), water (90%), swamp (89%), woodland (82%), grassland (81%), bushland (72%), and forest (60%). This indicates that forest experienced the highest conversion, with almost 26% of its total area converted to woodland and the rest to cultivated land (11%) and grassland (3%). The majority of bushland was converted to cultivated land (27%) and swamp (1%), while most of the grassland was converted to cultivated land (18%) and swamp (1%). Although the cultivated land did not change much, the largest share was gained from woodland 1605.06 km$^2$) followed by bushland (630.94 km$^2$), forest (114.27 km$^2$), grassland (111.80 km$^2$), and swamp (28.74 km$^2$). Between 2000 and 2016, cultivated land was converted to swamp, while water was converted to swamp and bushland respectively.

**Table 10.** Transition matrix showing LULC change in Wami Sub-catchment between 2000–2016.

| LULC | Unit | BL | WL | SWP | CL | SA | GL | WT | FR |
|------|------|------|------|------|------|------|------|------|------|
| BL | (km$^2$) | 1682.50 | 0 | 0 | 630.94 | 23.37 | 0 | 0 | 0 |
|  | % | 72 | 0 | 0 | 27 | 1 | 0 | 0 | 0 |
| WL | (km$^2$) | 0 | 7311.95 | 0 | 1605.06 | 0 | 0 | 0 | 0 |
|  | % | 0 | 82 | 0 | 18 | 0 | 0 | 0 | 0 |
| SWP | (km$^2$) | 0 | 2.87 | 255.75 | 28.74 | 0 | 0 | 0 | 0 |
|  | % | 0 | 1 | 89 | 10 | 0 | 0 | 0 | 0 |
| CL | (km$^2$) | 0 | 0 | 0 | 1142.96 | 35.35 | 0 | 0 | 0 |
|  | % | 0 | 0 | 0 | 97 | 3 | 0 | 0 | 0 |
| SA | (km$^2$) | 0 | 0 | 0 | 0 | 7.71 | 0 | 0 | 0 |
|  | % | 0 | 0 | 0 | 0 | 100 | 0 | 0 | 0 |
| GL | (km$^2$) | 0 | 0 | 6.21 | 111.80 | 0 | 503.11 | 0 | 0 |
|  | % | 0 | 0 | 1 | 18 | 0 | 81 | 0 | 0 |
| WT | (km$^2$) | 0.44 | 0.44 | 1.32 | 0 | 0 | 0 | 19.8 | 0 |
|  | % | 2 | 2 | 6 | 0 | 0 | 0 | 90 | 0 |
| FR | (km$^2$) | 0 | 270.08 | 0 | 114.27 | 0 | 31.16 | 0 | 623.27 |
|  | % | 0 | 26 | 0 | 11 | 0 | 3 | 0 | 60 |

BL–Bushland, WL–Woodland, SWP–Swamp, CL–Cultivated land, SA–Settlement area, GL–Grassland, WT–Water, FR–Forest. Note: Bold numbers on the diagonal represent unchanged LULC proportions from 2000 to 2016 and their corresponding percentages, while others are the areas changed from one class to another.

## 4. Discussion

The LULC changes in both sub-catchments have already been presented. The results for the study duration (2000–2016) on different classes of LULC indicate that most of the grassland, bushland, and woodland was intensively converted to cultivated land both upstream and downstream. The land use changes showed a marked impact of increase in population in the study area. The farmlands and grasslands were converted to settlements in Kinyasungwe Sub-catchment (upstream), while in Wami Sub-catchment (downstream), an increase in settlements resulted from conversion of the bushland and cultivated lands. The land use pattern observed agrees with other catchment areas in Tanzania, such as in Malagarasi, in which it was found that settlement and cultivation expansion were imminent [63].

The increasing trend of LULC change in the sub-catchments highlights the universal influence of economic forces in motivating principals for the anthropogenic alteration of land [64–66]. However, there is a variation in the pattern of LULC change of the different land use types. Furthermore, according to the findings, the decrease in bushland, woodland, swamp, and grassland was the consequence of the extension of cultivated land. This change of other LULC into cultivated land was supported by the change matrix tables, which show that from 2000 to 2016, 23% of bushland, 8% of woodland, 6% of the swamp, 18% of grassland, 2% of water, and 1% of the forest was converted into cultivated land upstream. However, the change matrix table for downstream shows that from 2000 to 2016, 27% of bushland, 18% of woodland, 9% of the swamp, 18% of grassland, and 11% of the forest converted into cultivated land, which implies there is a high demand of food production in the area. With the population growth experienced in 16 years, land demand for agriculture and settlement has increased. Therefore, the pressure on land resulted in the land use pattern observed [9]. The pressure and conversion of the natural areas has also been observed, especially woodland, bushland, and grassland upstream and natural forest downstream.

According to The International Union for Conservation of Nature (IUCN), Tanvir et al., and Butt et al. [64–66], the primary accelerator of forest decline in the catchment area is human activities. Findings from Wasige et al. [67] further corroborate these results, noting that the forest clearing occurring every year is mainly due to agriculture. A change in water area was also witnessed, decreasing both upstream by 68% and downstream by 10%. The increased change of water bodies into

swamp areas implies water shortage upstream. In general, the upstream was more affected by the reduction of land cover compared to downstream due to a significant increase in cultivated land to meet the high demand of food due to population increase, especially in Dar es Salaam City, resulting in a high demand of land for food production to fulfil necessities of life [68]. However, uncontrolled LULC change in the upstream sub-catchment can result in a reduction of the natural resources in the downstream sub-catchment, including water availability [68–70]. These downstream environmental degradations due to upstream influence are a significant cause of upstream–downstream conflicts [69].

## 5. Conclusions

A low-cost change analysis based on remote sensing imagery from different sensors made it possible to quantify and map the changing pattern in LULC in the Wami River Basin with emphasis on upstream and downstream changes. With a time-series of maps, change analysis can reveal the overall development of land distribution, including the detection of sites of different types of changes. Analyzing and mapping the trends of LULC changes within the Kinyasungwe and Wami Sub-catchments provides a basis for strategic planning, managing, and protection decision-making. However, the use of very high-resolution multispectral satellite imagery may offer more details of changes in the area. Based on the LULC analyses of Landsat data for the years 2000, 2006, 2011, and 2016, it was found that the LULC change trends varied significantly during the periods mentioned above. The results showed that in the period 2000–2016, most LULC were converted to cultivated land upstream (bushland 23%, grassland 18%, woodland 8%, swamp 6%, water 2%, and forest 1%) and downstream (bushland 27%, grassland 18% woodland 18%, forest 11%, and swamp 10%). This indicates that the expansion of cultivated land was the result of population growth, especially downstream, while the primary socioeconomic activity remains agriculture both upstream and downstream. In general, net gain and net loss were observed downstream, which indicates that it was more affected compared to upstream. Hence, proper management of the basin, including land-use planning, is required to avoid resource-use conflicts between upstream and downstream users.

**Author Contributions:** The two authors contributed in a substantial way to the manuscript. S.T. conceived and performed the research and wrote the manuscript. M.F.B. supervised the study at all stages and reviewed the manuscript. Both authors discussed the structure of the manuscript, read and approved the submitted manuscript.

**Funding:** This research was funded by the Ministry of Water, United Republic of Tanzania and the APC was funded by Dresden University of Technology (TU Dresden).

**Acknowledgments:** We acknowledge support from UNU-FLORES and TU Dresden. The first author would like to acknowledge the financial support from the Government of the United Republic of Tanzania.

**Conflicts of Interest:** The authors declare no conflict of interests.

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
