# Peer review of "Land-Use and Land-Cover (LULC) Change Detection in Wami River Basin, Tanzania"

_land, doi:10.3390/land8090136_

Round 1
Reviewer 1 Report
The ms was improved substantiallsubstantially. Please add aa better description of results with maps and further graphs. Check language usage. Please remove yellow at ref no. 16
Reviewer 2 Report
This is the third revision of the manuscript which looks much improved.
Tables 7 and 10 are not clear. Please provide the procedure how the quantities in these two tables are calculated.
The title of these two tables ‘LULC transition probabilities’ is also confusing because a ‘probability’ is not expressed in terms of sq km and percentages. More clear interpretation of these tables is needed in the results.
The whole manuscript should be carefully checked for typographical mistakes.
For example, on page 2 (para 2), ‘[27] 2010’ should be ‘[27] in 2010’.
The following sentence on page 2 para 3 should be revised: “Remote Sensing and GIS … over large areas”.
Reference number 50 0n page 13 should be corrected. The word ‘Remperature’ is in fact ‘Temperate’. The sequence of authors in this reference is also wrong, which should be corrected.
Author Response
Please see the attachment

This manuscript is a resubmission of an earlier submission. The following is a list of the peer review reports and author responses from that submission.
Round 1
Reviewer 1 Report
This re-submission of the paper reports the temporal changes in land use / land cover classes in the two sub-catchments of in Tanzania. The images from the Landsat series were used for the classification.
In regard to classification algorithm, it is puzzling that this version of paper mentions ‘Random Forest Classifier’ has been used but the previous version mentioned ‘Maximum Likelihood Algorithm’ was used, whereas the results remain the same.
There is a serious concern that certain questions that were raised in the previous version of the paper have not been adequately addressed.
The big issue which has not been addressed is about the claim that “in the period 2000-2016, 58% and 80% of other land-use/land-cover was converted to cultivated land at Kinyasungwe sub-catchment and Wami sub-catchment respectively”. According to the data provided in Tables 5 and 6, these percentages are incorrect. These should be 15.3 and 18.7. The answer provided by authors to this question is very confusing and sounds illogical.
Table 1 provides the acquisition dates of the satellite images used in this study. Accuracy of the dates needs to be checked. For example, the acquisition date of the Landsat-8 scene 167/64 is mentioned as 26 Feb 2016, which is not correct. In fact, as per USGS records, the two acquisitions of this particular scene in that month were on 02 and 18 Feb 2016, and none on 26 Feb.
On page 4, the phrase “…similar spectral resolution, i.e. 30 metres” defies the basic concept of word ‘spectral’. This word should be ‘spatial’. However, the question remains whether the signature sets of specified classes were separately collected for each of the three sensors (OLI/TIRS, ETM+ and TM) or not. If not, explanation is required on how the inter-sensor differences were handled.
Tables 4, 9 and 10 have not been cited in the main text though authors claim that all tables have been cited.
There is still no explanation of the change detection tables (Tables 9 and 10) provided in the text.
Reviewer 2 Report
My impression is that this article is a mere description of land use changes over 16 years in a study area of Tanzania. With this content, I am not sure the manuscript represents a sufficient contribution to Land. What is the novelty of your work? To achieve publication, the authors should better clarify the aims and scope of their work, elaborating a bit more the novelty and originality of their results. Why the study area has been selected? What is the value added to study your area for continental and global analysis? Please elaborate substantially on these key issue, with the final aim to give further value to your paper.